# Select-and-Sample for Spike-and-Slab Sparse Coding

**Abdul-Saboor Sheikh**
Technical University of Berlin, Germany,
and Cluster of Excellence Hearing4all
University of Oldenburg, Germany,
and SAP Innovation Center Network, Berlin
sheikh.abdulsaboor@gmail.com

**Jörg Lücke**
Research Center Neurosensory Science
and Cluster of Excellence Hearing4all
and Dept. of Medical Physics and Acoustics
University of Oldenburg, Germany
joerg.luecke@uol.de

## Abstract

Probabilistic inference serves as a popular model for neural processing. It is still unclear, however, how approximate probabilistic inference can be accurate and scalable to very high-dimensional continuous latent spaces. Especially as typical posteriors for sensory data can be expected to exhibit complex latent dependencies including multiple modes. Here, we study an approach that can efficiently be scaled while maintaining a richly structured posterior approximation under these conditions. As example model we use spike-and-slab sparse coding for V1 processing, and combine latent subspace selection with Gibbs sampling (select-and-sample). Unlike factored variational approaches, the method can maintain large numbers of posterior modes and complex latent dependencies. Unlike pure sampling, the method is scalable to very high-dimensional latent spaces. Among all sparse coding approaches with non-trivial posterior approximations (MAP or ICA-like models), we report the largest-scale results. In applications we firstly verify the approach by showing competitiveness in standard denoising benchmarks. Secondly, we use its scalability to, for the first time, study highly-overcomplete settings for V1 encoding using sophisticated posterior representations. More generally, our study shows that very accurate probabilistic inference for multi-modal posteriors with complex dependencies is tractable, functionally desirable and consistent with models for neural inference.

## 1 Introduction

The sensory data that enters our brain through our sensors has a high intrinsic dimensionality and it is complex and ambiguous. Image patches or small snippets of sound, for instance, often do not contain sufficient information to identify edges or phonemes with high degrees of certainty. Probabilistic models are therefore very well suited to maintain uncertainty encodings. Given an image patch, for instance, high probabilities for an edge in one location impacts the probabilities for other components resulting in complex dependencies commonly known as "explaining-away" effects. Such dependencies in general include (anti-)correlations, higher-order dependencies and multiple posterior modes (i.e., alternative interpretations of a patch). Furthermore, sensory data is typically composed of many different elementary constituents (e.g., an image patch contains some of a potentially very large number of components) resulting in sparse coding models aiming at increasing overcompleteness [1]. If sensory data gives rise to complex posterior dependencies and has high intrinsic dimensionality, how can we study inference and learning in such settings? To date most studies, e.g. of V1 encoding models, have avoided the treatment of complex latent dependencies by assuming standard sparse models with Laplace priors [2, 3, 1]; high-dimensional problems can then be addressed by applying maximum a-posteriori (MAP) approximations for the resulting mono-modal posteriors. Other scalable approaches such as independent component analysis (ICA) or singular value decomposition (K-SVD) [4, 5] do not encode for data uncertainty, which

avoids posterior estimations altogether. For advanced data models, which we expect to be required, e.g., for visual data, neither MAP nor a non-probabilistic treatment can be expected to be sufficient. It was, for example, in a number of studies shown that sparse coding models with more flexible spike-and-slab priors are (A) more closely aligned with the true generative process e.g. for images, and are (B) resulting in improved functional performance [6, 7]. Spike-and-slab priors do, however, result in posteriors with complex dependencies including many modes [8, 7]. Inference w.r.t. to spike-and-slab sparse coding is therefore well suited to, in general, study efficient inference and learning with complex posteriors in high-dimensions. Results for spike-and-slab sparse coding are, furthermore, of direct interest for other important models such as hierarchical communities of experts [9], deep Boltzmann Machines (see [6]), or convolutional neural networks [10]. Also for these typically deep systems very high-dimensional inference and learning is of crucial importance.

So far, intractable inference for spike-and-slab sparse coding was approximated using sampling or factored variational approaches. While sampling approaches can in principle model any dependencies including multiple modes, they have been found challenging to train at scale, with the largest scale applications going to few hundred of latents [11, 12]. Compared to sampling, approaches using factored variational approximations can not model as complex posterior dependencies because they assume posterior independence (no correlations etc), however, they can capture multiple modes and are scalable to several hundreds up to thousands of latents [8, 6]. In this work we combine the accuracy of sampling approaches and the scalability of variational approaches by applying select-and-sample [13] to scale spike-and-slab sparse coding to very high latent dimensions. In contrast to using a factored approximation, we here select low dimensional subspaces of the continuous hidden space, and then apply sampling to approximate posteriors within these lower dimensional spaces.

## 2 The Spike-and-Slab Sparse Coding Model and Parameter Optimization

The spike-and-slab sparse coding model (see [8, 6] and citations therein) used for our study assumes a Bernoulli prior over all $H$ components of the the binary latent vector $\vec{b} \in \{0, 1\}^H$, with a Gaussian prior (the 'slab') for the continuous latent vector $\vec{z} \in \mathbb{R}^H$:

$$p(\vec{b} \,|\, \Theta) = \prod_h \pi^{b_h} (1 - \pi)^{1-b_h}, \quad p(\vec{z} \,|\, \Theta) = \prod_h \mathcal{N}(z_h; \mu_h, \psi_h^2), \tag{1}$$

where $\pi$ defines the probability of $b_h$ being equal to one and where $\vec{\mu} \in \mathbb{R}^H$ and $\vec{\psi} \in \mathbb{R}^H$ parameterize the Gaussian slab. A spike-and-slab hidden variable $\vec{s} \in \mathbb{R}^H$ is then generated by a pointwise multiplication: $\vec{s} = (\vec{b} \odot \vec{z})$, i.e., $s_h = b_h z_h$. Given the hidden variable $\vec{s}$, we follow standard sparse coding by linearly superimposing a set of latent components (i.e., $W\vec{s} = \sum_h \vec{W}_h s_h$) to initialize the mean of a Gaussian noise model:

$$p(\vec{y} \,|\, \vec{s}, \Theta) = \prod_d \mathcal{N}(y_d; \sum_h W_{dh} s_h, \sigma_d^2), \tag{2}$$

which then generates us the observed data $\vec{y} \in \mathbb{R}^D$. Here the columns of the matrix $W \in \mathbb{R}^{D \times H}$ are each a latent component $\vec{W}_h$ that is associated with a spike-and-slab latent variable $s_h$. We use $\vec{\sigma} \in \mathbb{R}^D$ to parameterize the observation noise. The parameters of the generative model (1) to (2) are together denoted by $\Theta = (\pi, \vec{\mu}, \vec{\psi}, W, \vec{\sigma})$. To find the values of $\Theta$, we seek to maximize the data likelihood $\mathcal{L} = \prod_{n=1}^N p(\vec{y}^{(n)} \,|\, \Theta)$ under the spike-and-slab data model and given a set of $N$ data points $\{\vec{y}^{(n)}\}_{n=1,\ldots,N}$. To derive a learning algorithm, we apply expectation maximization (EM) in its free-energy formulation. In our case the free-energy is given by:

$$\mathcal{F}(q, \Theta) = \sum_{n=1}^N \left\langle \log p(\vec{y}^{(n)}, \vec{s} \,|\, \Theta) \right\rangle_n + H(q^{(n)}), \text{ where } \langle f(\vec{s}) \rangle_n = \int q^{(n)}(\vec{s}) \, f(\vec{s}) d\vec{s}$$

is the expectation under $q^{(n)}$, a distribution over the latent space and $H(\cdot)$ denotes the Shannon entropy. Given the free-energy, the parameter updates are canonically derived by setting the partial derivatives of $\mathcal{F}(q, \Theta)$ w.r.t. the parameters to zero. For the spike-and-slab sparse coding model (1) and (2), we obtain (similar to [8, 6, 7]) the following closed-form M-step equations:

$$\pi = \frac{1}{NH} \sum_n \langle |\vec{b}| \rangle_n, \quad \psi_h^2 = \frac{\sum_n \langle (s_h - \mu_h b_h)^2 \rangle_n}{\sum_n \langle b_h \rangle_n}, \tag{3}$$

$$W = \frac{\sum_n \vec{y}^{(n)} \langle \vec{s} \rangle_n^{\mathrm{T}}}{\sum_n \langle \vec{s}\vec{s}^{\mathrm{T}} \rangle_n}, \quad \vec{\mu} = \frac{\sum_n \langle \vec{s} \rangle_n}{\sum_n \langle \vec{b} \rangle_n}, \text{ and } \sigma_d^2 = \frac{1}{N} \sum_n \langle (\sum_h W_{dh} s_h - y_d^{(n)})^2 \rangle_n \tag{4}$$

with $s_h = b_h z_h$ and $|\vec{x}| = \sum_h x_h$ as defined above.

# 3   Approximate Inference With Select-and-Sample

The optimal choices for the distributions $q^{(n)}(\vec{s})$ for the expectations in (3) and (4) are the posteriors $p(\vec{s} | \vec{y}^{(n)}, \Theta)$, but neither the posteriors nor their corresponding expectation values are computationally tractable for high dimensions. However, a crucial observation that we exploit in our work is that for observed data such as natural sensory input or data generated by a sparse coding model, the activity of latent components (or causes) can be expected to be concentrated in low-dimensional subspaces. In other words, for a given observed data point, all except for a very small fraction of the latent components can be assumed to be non-causal or irrelevant, hence the corresponding latent space can be neglected for the integration over $\vec{s}$. For a sparse initiation (i.e., $\pi \ll 1$) of the spike-and-slab model (1) to (2), we consider such low dimensional subspaces to be spanned by a few (approximately $\pi H$) of the $H$ latent space coordinates. If we denote by $\mathcal{J}^{(n)}$ the subspace containing the large majority of posterior mass for a given data point $\vec{y}^{(n)}$, an approximation to $p(\vec{s} | \vec{y}^{(n)}, \Theta)$ is then given by the following truncated distribution:

$$q^{(n)}(\vec{s}; \Theta) = \frac{p(\vec{s} | \vec{y}^{(n)}, \Theta)}{\int_{\vec{s}' \in \mathcal{J}^{(n)}} p(\vec{s}' | \vec{y}^{(n)}, \Theta) \, d\vec{s}'} \, \delta(\vec{s} \in \mathcal{J}^{(n)}), \tag{5}$$

where $\delta(\vec{s} \in \mathcal{J}^{(n)})$ is an indicator function, taking the value $\delta(\vec{s} \in \mathcal{J}^{(n)}) = 1$ only if $\vec{s} \in \mathcal{J}^{(n)}$ and zero otherwise. Truncated approximations have previously been shown to work efficiently and accurately for challenging data models [14, 15, 16]. Latents were restricted to be binary, however, and scalability was previously limited by the combinatorics within the selected latent subsets. For our aim of very large scale applications, we therefore apply the select-and-sample approach [13] and use a sampling approximation that operates within the subspaces $\mathcal{J}^{(n)}$. Unlike [13] who used binary latents, we here apply the approximation to the continuous latent space of spike-and-slab sparse coding. Formally, this means that we first use the posterior approximation $q^{(n)}(\vec{s})$ in Eqn. 5 and then approximate the expectation values w.r.t. $q^{(n)}(\vec{s})$ using sampling (see illustration of Alg. 1):

$$\langle f(\vec{s}) \rangle_n = \int q^{(n)}(\vec{s}) f(\vec{s}) \, d\vec{s} \approx \frac{1}{M} \sum_{m=1}^{M} f(\vec{s}^{(m)}), \text{where } \vec{s}^{(m)} \sim q^{(n)}(\vec{s}), \tag{6}$$

$M$ is the number of samples and $f(\vec{s})$ can be any argument of the expectation values in (3) and (4).

It remains to be shown how difficult sampling from $q^{(n)}(\vec{s})$ is compared to directly sampling form the full posterior $p(\vec{s} | \vec{y}^{(n)}, \Theta)$. The index function $\delta(\vec{s} \in \mathcal{J}^{(n)})$ means that we can clamp all values of $\vec{s}$ to zero but we have to answer the question how the remaining $s_h$ are sampled. A closer analysis of the problem shows that the distribution to sample in the reduced space is given by the posterior w.r.t. a truncated generative model. To show this, let us first introduce some notation: Let us denote by $\mathcal{I}$ a subset of the indices of the latent variables $\vec{s}$, i.e., $\mathcal{I} \subseteq \{1, \ldots, H\}$, and let us use $H \backslash \mathcal{I}$ as an abbreviation for $\{1, \ldots, H\} \backslash \mathcal{I}$. The vector $\vec{s}_{\mathcal{I}}$ w.r.t. $\mathcal{I}$ is then, as customary, a vector in $\mathbb{R}^{|\mathcal{I}|}$ defined by those entries $s_h$ with $h \in \mathcal{I}$. In analogy, we take a matrix $W_{\mathcal{I}} \in \mathbb{R}^{D \times |\mathcal{I}|}$ to be defined by row vectors $(\vec{w}_d^T)_{\mathcal{I}}$ where $\vec{w}_d^T$ are the row vectors of $W \in \mathbb{R}^{D \times H}$.

**Proposition 1.** Consider the spike-and-slab generative model (1) to (2) with parameters $\Theta$, and let $\Theta_{\mathcal{I}^{(n)}} = (\pi, \vec{\mu}_{\mathcal{I}^{(n)}}, \vec{\psi}_{\mathcal{I}^{(n)}}, W_{\mathcal{I}^{(n)}}, \vec{\sigma})$ be the parameters of a truncated spike-and-slab model with $H' = \dim(\mathcal{I}^{(n)})$ dimensional latent space. Then it applies that sampling from the truncated distribution in (5) is equivalent to sampling from the posterior $p(\vec{s}_{\mathcal{I}^{(n)}} | \vec{y}^{(n)}, \Theta_{\mathcal{I}^{(n)}})$ of the truncated spike-and-slab model, while all values $s_h$ with $h \notin \mathcal{I}^{(n)}$ are clamped to zero.

**Proof.** If $\mathcal{I}^{(n)}$ denotes the indices of those latents $s_h$ that span the subspace in which the posterior mass of $p(\vec{s} | \vec{y}^{(n)}, \Theta)$ is concentrated, then these subsets are given by $\mathcal{J}^{(n)} = \{\vec{s} \in \mathbb{R}^H | \vec{s}_{H \backslash \mathcal{I}^{(n)}} = \vec{0}\}$, i.e., $\delta(\vec{s} \in \mathcal{J}^{(n)})$ can be rewritten as $\prod_{h \notin \mathcal{I}^{(n)}} \delta(s_h = 0)$. Considering (5), we can therefore set the corresponding values $\vec{s}_{H \backslash \mathcal{I}^{(n)}} = \vec{0}$. We now drop the

superscript $n$ for readability and first derive:

$$p(\vec{s}_{\mathcal{I}}, \vec{s}_{H \setminus \mathcal{I}} = \vec{0}, \vec{y} \,|\, \Theta)$$

$$= \mathcal{N}(\vec{y}; W_{\mathcal{I}} \vec{s}_{\mathcal{I}} + W_{H \setminus \mathcal{I}} \vec{0}, \vec{\sigma}) \Big( \prod_{h \in \mathcal{I}} \mathrm{Bern}(b_h; \pi) \mathcal{N}(z_h; \mu_h, \psi_h) \Big) \Big( \prod_{h \notin \mathcal{I}} \mathrm{Bern}(b_h = 0; \pi) \mathcal{N}(z_h; \mu_h, \psi_h) \Big)$$

$$= p(\vec{s}_{\mathcal{I}}, \vec{y} \,|\, \Theta_{\mathcal{I}}) \, \mathcal{U}(\vec{s}_{H \setminus \mathcal{I}} = \vec{0}, \Theta) \quad \text{with} \quad \mathcal{U}(\vec{s}_{H \setminus \mathcal{I}}, \Theta) = p(\vec{s}_{H \setminus \mathcal{I}} \,|\, \Theta_{H \setminus \mathcal{I}}),$$

i.e., the joint with $\vec{s}_{H \setminus \mathcal{I}} = \vec{0}$ is given by the joint of the truncated model multiplied by a term not depending on $\vec{s}_{\mathcal{I}}$ such that:

$$q(\vec{s}; \Theta) = \frac{p(\vec{s}_{\mathcal{I}}, \vec{s}_{H \setminus \mathcal{I}} = \vec{0}, \vec{y} \,|\, \Theta) \, \delta(\vec{s} \in \mathcal{J})}{\displaystyle\int_{\vec{s}' \in \mathcal{J}} p(\vec{s}'_{\mathcal{I}}, \vec{s}'_{H \setminus \mathcal{I}} = \vec{0}, \vec{y} \,|\, \Theta) \, \mathrm{d}\vec{s}'} = \frac{p(\vec{s}_{\mathcal{I}}, \vec{y} \,|\, \Theta_{\mathcal{I}}) \, \mathcal{U}(\vec{s}_{H \setminus \mathcal{I}} = \vec{0}, \Theta) \, \delta(\vec{s} \in \mathcal{J})}{\displaystyle\int_{\vec{s}' \in \mathcal{J}} p(\vec{s}'_{\mathcal{I}}, \vec{y} \,|\, \Theta_{\mathcal{I}}) \, \mathrm{d}\vec{s}' \, \mathcal{U}(\vec{s}_{H \setminus \mathcal{I}} = \vec{0}, \Theta)}$$

$$= \frac{p(\vec{s}_{\mathcal{I}}, \vec{y} \,|\, \Theta_{\mathcal{I}})}{\displaystyle\int_{\vec{s}'} p(\vec{s}'_{\mathcal{I}}, \vec{y} \,|\, \Theta_{\mathcal{I}}) \, \mathrm{d}\vec{s}'_{\mathcal{I}}} \prod_{h \notin \mathcal{I}} \delta(s_h = 0) = p(\vec{s}_{\mathcal{I}} \,|\, \vec{y}, \Theta_{\mathcal{I}}) \prod_{h \notin \mathcal{I}} \delta(s_h = 0). \qquad (7)$$

$\square$

Following the proof, Proposition 1 applies for any generative model $p(\vec{s}, \vec{y} \,|\, \Theta)$ for which applies $p(\vec{s}_{\mathcal{I}}, \vec{s}_{H \setminus \mathcal{I}} = \vec{0}, \vec{y} \,|\, \Theta) = p(\vec{s}_{\mathcal{I}}, \vec{y} \,|\, \Theta_{\mathcal{I}}) \, \mathcal{U}(\vec{s}_{H \setminus \mathcal{I}} = \vec{0}, \vec{y}, \Theta)$. This includes a large class of models such as linear and non-linear spike-and-slab models, and potentially hierarchical models such as SBNs. Proposition 1 does not apply in general, however (we exploit specific model properties).

*Sampling.* In previous work [7], posteriors for spike-and-slab sparse coding have been evaluated exhaustively within selected $\mathcal{I}^{(n)}$ which resulted in scalability to be strongly limited by the dimensionality of $\mathcal{I}^{(n)}$. Based on Proposition 1, we can now overcome this bottleneck by using sampling approximations within the subspaces $\mathcal{J}^{(n)}$, and we have shown that such sampling is equivalent to sampling w.r.t. to a much lower dimensional spike-and-slab model. The dimensionality of $\mathcal{J}^{(n)}$ is still non-trivial, however, and we use a Markov chain Monte Carlo (MCMC) approach, namely Gibbs sampling for efficient scalability. Following Proposition 1 we derive a sampler for the spike-and-slab model (1) to (2) and later apply it for the needed (low) dimensionality.

While the result of sampling from posteriors of truncated models applies for a broad class of spike-and-slab models (Proposition 1), we can here exploit a further specific property of the model (1) to (2). As has previously been observed and exploited in different contexts [8, 12, 17], the Gaussian slab and the Gaussian noise model can be combined using Gaussian identities such that integrals over the continuous latents $\vec{z}$ are solvable analytically. Here we can use this observation for the derivation of a Gibbs sampler. For this we first devise a latent variable Markov chain such that its target density is given by the following conditional posterior distribution:

$$p(s_h | \vec{s}_{H \setminus h}, \vec{y}, \theta) \propto p(s_h | \theta) \prod_d p(y_d | s_h, \vec{s}_{H \setminus h}, \theta)$$

$$= \Big( (1 - \pi) \, \tilde{\delta}(s_h) + \pi \, \mathcal{N}(s_h; \mu_h, \psi_h^2) \Big) \prod_d \mathcal{N}(s_h; \nu_d, \varphi_d^2), \qquad (8)$$

where $\tilde{\delta}(.)$ is the Dirac delta to represent the spike at zero and where $\nu_d = (y_d - \sum_{h' \setminus h} W_{dh'} s_{h'})/W_{dh}$ and $\varphi_d^2 = \sigma_d^2 / W_{dh}^2$. Using Gaussian identities we obtain:

$$p(s_h | \vec{s}_{H \setminus h}, \vec{y}, \theta) \propto \Big( (1 - \pi) \mathcal{N}(s_h; \upsilon, \phi^2) \, \tilde{\delta}(s_h) + \pi \, \mathcal{N}(s_h; \tau, \omega^2) \Big), \qquad (9)$$

where $\upsilon = \phi^2 \sum_d \nu_d / \varphi_d^2$ and $\phi^2 = (\sum_d 1/\varphi_d^2)^{-1}$, whereas $\tau = \omega^2 \, (\upsilon/\phi^2 + \mu_h/\psi_h^2)$ and $\omega^2 = (1/\phi^2 + 1/\psi_h^2)^{-1}$. We can observe that the conditional posterior (9) of $s_h$ retains the form of a spike-and-slab distribution. We can therefore simply compute the cumulative distribution function (CDF) of (9) to simulate $s_h$ from the exact conditional distribution ($s_h \sim p(s_h | \vec{s}_{H \setminus h}, \vec{y}, \theta)$) by means of inverse transform sampling.

*Selecting.* The Gibbs sampler can now be applied to generate posterior samples for a truncated spike-and-slab model (defined using parameters $\Theta_{\mathcal{I}^{(n)}}$). We also obtain a valid approximation, of course, without selection ($\mathcal{I} = \{1, \ldots, H\}$) but MCMC samplers in very high dimensional spaces

**Algorithm 1:** Select-and-sample for spike-and-slab sparse coding (S5C)

init $\Theta$;
**repeat**
    **for** $(n = 1, ..., N)$ **do**
        **for** $(h = 1, ..., H)$ **do**
            compute $\mathcal{S}_h(\vec{y}^{(n)})$ as in (10);
        define $\mathcal{I}^{(n)}$ as in (11);
        **for** $(m = 1, \ldots, M)$ **do**
            draw $\vec{s}^{(m)}_{\mathcal{I}^{(n)}} \sim p(\vec{s}_{\mathcal{I}^{(n)}} \,|\, \vec{y}^{(n)}, \Theta_{\mathcal{I}^{(n)}})$ using (9);
        compute $\langle f(\vec{s}) \rangle_n = \frac{2}{M} \sum_{m=\frac{M}{2}+1}^{M} f(\vec{s}^{(m)})$;
    compute M-step with arguments $f(\vec{s})$ as in (3) and (4);
**until** *(until $\Theta$ have converged)*;

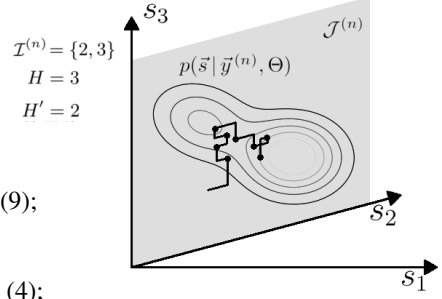

Illustration of general application.

with complex posterior structure are known to be challenging (convergence to target distributions can be very slow). The problems typically increase superlinearly with hidden dimensionality but for intermediate dimensions, a Gibbs sampler can be very fast and accurate. By using subspaces $\mathcal{J}^{(n)}$ with intermediate dimensionality, therefore, results in very efficient and accurate sampling approximations within these spaces. An overall very accurate approximation is then obtained if the subspaces are well selected and if they do contain the large majority of posterior mass. By using exact EM it was indeed previously shown for spike-and-slab sparse coding [7] that almost all posterior mass, e.g., for naturally mixed sound sources, is concentrated in collections of low-dimensional subspaces (also compare [18]). To define a subspace $\mathcal{J}^{(n)}$ given a data point $\vec{y}^{(n)}$, we follow earlier approaches [15, 14] and first define an efficiently computable selection function to choose those latents that are the most likely to have generated the data point. We use the selection function in [7] which is given by:

$$\mathcal{S}_h(\vec{y}^{(n)}, \Theta) = \sum_d \mathcal{N}(\vec{y}^{(n)}_d; W_{dh}\mu_h, \sigma_d + W^2_{dh}/\psi_h) \propto p(\vec{y}^{(n)} \,|\, \vec{b} = \vec{b}_h, \Theta), \qquad (10)$$

where $\vec{b}_h$ represents a singleton state with only component $h$ being equal to one. The subsets are then defined as follows:

$$\mathcal{I}^{(n)} \text{ is the set of } H' \text{ indices such that } \forall h \in \mathcal{I}^{(n)} \ \forall h' \notin \mathcal{I}^{(n)} : \ \mathcal{S}_h(\vec{y}^{(n)}, \Theta) > \mathcal{S}_{h'}(\vec{y}^{(n)}, \Theta). \ (11)$$

We then use $\mathcal{J}^{(n)} = \{\vec{s} \,|\, \vec{s}_{H \setminus \mathcal{I}^{(n)}} = \vec{0}\}$ as above. In contrast to previous approaches with $H'$ typically $< 10$, $H'$ can be chosen relatively large here because the Gibbs sampler is still very efficient and precise for $H' > 10$ (we will go up $H' = 40$).

By combining selection procedure and the Gibbs sampler using Proposition 1, we obtain the efficient approximate EM algorithm summarized in Alg. 1. It will be referred to as S5C (see Alg. 1 caption). Note that we will, for all experiments, always discard the first half of the drawn samples as burn-in.

## 4  Numerical Experiments

In all the experiments, the initial values of $\pi$ were drawn from a uniform distribution on the interval $[0.1, 0.5]$ (i.e., intermediately sparse), $\vec{\mu}$ was initialized with normally distributed random values, $\psi_h$ was set to 1 and $\sigma_d$ was initialized with the standard deviation of $y_d$. The elements of $W$ were iid drawn from a normal distribution with zero mean and a standard deviation of 5.0. We used a multi-core parallelized implementation and executed the algorithm on up to 1000 CPU cores.

**Verification of functional accuracy.** We first investigate the accuracy and convergence properties of our method on ground-truth data which was generated by the spike-and-slab data model (1) and (2) itself. We used $H = 10$ hidden variables and $D = 5 \times 5$ and generative fields $\vec{W}_h$ in the form of five horizontal and five vertical bars. As is customary for such bars like data (e.g., [15] and cites therein) we take each field to contribute to a data point with probability $\pi = \frac{2}{H}$. We then randomly make each of the 5 vertical and 5 horizontal bars positive or negative by assigning them a value of 5

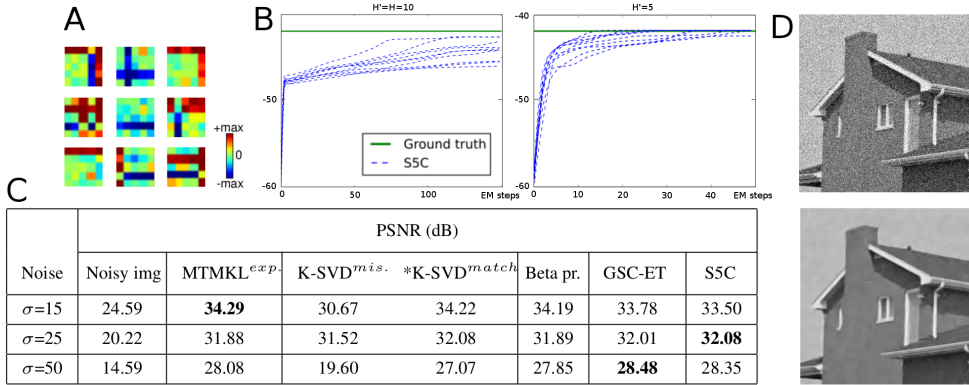

Figure 1: Functional accuracy of S5C. **A** Artificial ground-truth data. **B** Likelihoods during learning (Alg. 1) for different $H'$. **C** Denoising performance of S5C on the 'house' benchmark as used for other methods (MTMKL [8], K-SVD [4], Beta process [11] and GSC-ET [7]). Bold values highlight the best performing algorithm. $^*$Value not bold-faced as noise variance is assumed known a-priori[4]. **D** Top: Noisy image with $\sigma = 25$. Bottom: State-of-the art denoising result after S5C was applied.

or $-5$, while the non-bar pixels are assigned zero value. The parameters of the latent slabs $\mu_h$ and $\psi_h$ are set to 0.0 and 1.0, respectively, and we set the observation noise to $\sigma_d = 1.0$. We generate $N = 5000$ data points with this setting (see Fig. 1A for examples).

We apply the S5C algorithm (Alg. 1) with $H = 10$ latents and $M = 40$ samples per data point and use two settings for preselection: (A) no preselection ($H' = H = 10$) and (B) subspace preselection using $H' = 5$. We did ten runs per setting using different initializations per run as above. For setting (A), i.e. pure Gibbs sampling, the algorithm recovered, after 150 EM iterations, the generating bars in 2 of the 10 runs. For setting (B) convergence was faster and in 9 of the 10 runs all bars were recovered after 50 EM iterations. Fig. 1B shows for all 20 runs likelihoods during learning (which are still tractable for $H = 10$). These empirical results show the same effect for a continuous latent variable model as was previously reported for non-continuous latents [19, 20]: preselection helps avoiding local optima (presumably because poor non-sparse solutions are destabilized using subspace selection).

After having verified the functioning of S5C on artificial data, we turned to verifying the approach on a denoising benchmark, which is standard for sparse coding. We applied S5C using a noisy "house" image [following 11, 4, 8, 7]. We used three different levels of added Gaussian noise ($\sigma = 15, 25, 50$). For each setting we extract $8 \times 8$ patches from $256 \times 256$ noisy image, visiting a whole grid of $250 \times 250$ pixels by shifting (vertically and horizontally) 1 pixel at a time. In total we obtained $N = 62,001$ overlapping image patches as data points. We applied the S5C algorithm with $H = 256$, select subspaces with $H' = 40$ and used $M = 100$ samples per subspace. Fig. 1C,D show the obtained results and a comparison to alternative approaches. As can be observed, S5C is competitive to other approaches and results in higher peak signal-to-noise ratios (PSNRs) (see [7] for details) than, e.g., K-SVD or factored variational EM approaches (MTMKL) for $\sigma = 25$ and 50. Even though S5C uses the additional sampling approximation in the selected subspaces, it is also competitive to ET-GSC [7], which is less efficient as it sums exhaustively within subspaces. For $\sigma = 25$ S5C even outperforms ET-GSC presumably because S5C allows for selecting larger subspaces. In general we observed increased improvement with the number of samples, but improvements with $H$ saturated after about $H = 256$.

**Large-scale application and V1 encoding.** Since sparse coding was first suggested as coding model for primary visual cortex [21], a main goal has been its application to very high latent dimensions because V1 is believed to be highly overcomplete [1]. Furthermore, for very large hidden dimensions, non-standard generative fields were observed [1], a finding which is of significant relevance for the ongoing debate of how and where increasingly complex structures in the visual system may be processed. Here we applied S5C with $H = 10\,000$ hidden dimensions to demonstrate scalability of the method, and to study highly-overcomplete V1 encoding based on a posterior approximation capturing rich structure. For our application we used the standard van Hateren database [22], extracted $N = 10^6$ image patches of size $16 \times 16$, and applied pseudo-whitening following [21]. We applied

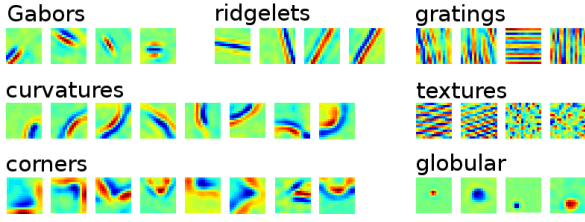

Gabors      ridgelets      gratings

curvatures      textures

corners      globular

Figure 2: Selection of different types of generative fields as learned by S5C using $H = 10,000$ latent dimensions (see Suppl. for all fields). Gabor-like fields are the most frequent type (Gabors, ridgelets, gratings), followed by globular fields, curved fields and corner-like fields. We also observed textures other than gratings. Gabors, curved and corner fields were almost all among the 30% most frequently activated fields. Ridgelets, globular fields and gratings were typically among the 30-80% most used fields.

S5C for 50 EM iterations to the data using $H' = 20$ dimensional subspaces and $M = 50$ samples per data point. After learning we observed a large number of generative fields specialized to image components. Like in recent large-scale applications of standard sparse coding [1] we found fields that did not specialize (about 1% in [1] and about 12% for S5C). The higher percentage for S5C may be due to the five-fold higher dimensionality used here. For the fields specialized to components, we observed a large number of Gabor-like fields including ridgelets and gratings (names follow [1]). Furthermore, we observed globular fields that have been observed experimentally [23] and are subject of a number of recent theoretical studies (e.g., [14, 3]). Notably, we also observed a number of curved fields and fields sensitive to corner-like structures (Fig. 2 shows some examples). Curved fields have so far only been described to emerge from sparse coding once before [1] and for convolutional sparse coding in two cases [24, 25] (to the knowledge of the authors) but have been suggested for technical applications much earlier [26] (a link that was not made, so far). Corner-like structures have previously not been observed for sparse coding presumably because of lower dimensional latent spaces (also not in [1] but compare convolutional extensions [24, 16, 25]). The numbers of curved (a few hundred) and corner-like fields (a few tens) are small but we almost exclusively find those fields among the 20% most frequently used fields (we order according to average approx. posterior, see supplement). Neural responses to corner-like sensitivities are typically associated with higher-level processing in the visual pathway. Our results may be evidence for such structures to emerge together, e.g., with Gabors for very large latent dimensionality (as expected for V1). In general, the statistics of generative field shapes can be influenced by many factors including preprocessing details, sparsity, local optima or details of the learning algorithms. However, because of the applied approximation, S5C can avoid the for MAP based approaches required choice of the sparsity penalty [1]. Instead we statistically infer the sparsity level which is well interpretable for hard sparsity, and which corresponds for our application to $H\pi = 6.2$ components per patch (also compare [14, 20]). In the supplement we provide the full set of the $H = 10\,000$ learned generative fields.

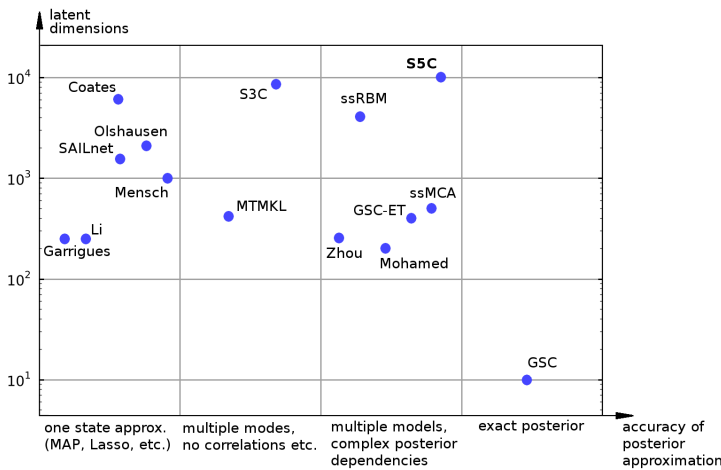

Figure 3: The y-axis shows the highest reported latent dimensionality for different sparse coding algorithms (cont. latents), and the x-axis the accuracy of posterior approximations. Within each column, entries are ordered (left-to-right) w.r.t. the publication year. 1st column: Sparse coding systems using one latent state for inference (eg., MAP-like [27, 28, 1] or SAILnet [3] or K-SVD [4, 5]). 2nd: Approximate posteriors in the form of factored variational distributions that can capture multiple modes but assume posterior independence among the latents $s_h$ (MTMKL [8], S3C [6]). 3rd: Sampling based approximations [11, 12] and truncated approximations (ssMCA [20], GSC-ET [7]) that capture multiple posterior modes and complex latent dependencies. Following [6] we also included ssRBM for comparison. 4th: Full posterior with exact EM [17].

# 5 Discussion

In this study we have applied a select-and-sample approach [13] to derive and study an approximate EM algorithm applicable to models with very large-scale latent spaces. Select-and-sample combines sampling with subspace preselection [15, 14] and has previously been applied as model for neural inference using binary latents [13]. Furthermore, it has been used to overcome analytical intractabilities of a non-linear sparse coding model [20]. Here, we for the first time apply select-and-sample to scale a standard linear sparse coding model with spike-and-slab prior up to very large hidden dimensions. Spike-and-slab sparse coding is hereby not only more expressive than standard Laplace or binary priors [8, 12, 7, 20] but results in properties that we can exploit for our approximation. We have thus analytically shown (Proposition 1) that select-and-sample is applicable to a large class of models with *hard* sparsity (giving justification also to earlier applications [20]).

Empirically, we have, firstly, shown that select-and-sample for spike-and-slab sparse coding (S5C) maintains the functional competitiveness of alternative approaches (Fig. 1). Secondly, we demonstrated efficiency by scaling S5C up to very high-dimensional latent spaces (we go up to $10\,000$). For comparison, Fig. 3 shows the largest reported latent spaces of different sparse coding approaches depending on the posterior structure that can be captured. Non-probabilistic approaches (e.g., K-SVD [4, 5]) are known to scale relatively well, and, likewise, approaches using MAP approximations [2, 3, 1] have been shown to be applicable to large scales. None of these approaches captures posterior dependencies or multiple posterior modes given a data point, however. Factored variational approaches can be scaled to very high-dimensional latent spaces and can capture multiple posterior modes. No latent dependencies in the posterior are modeled, however, which has previously been reported to result in disadvantageous behavior (e.g. [29, 7]). In contrast to MAP-based or factored approaches, sampling approaches can model both multiple posterior modes and complex latent dependencies. Some models hereby additionally include a more Bayesian treatment of parameters [11, 12] (also compare [8]) which can be considered more general than approaches followed in other work (see Fig. 3). The scalability of sampling based approaches has been limited, however. Among those models capturing the crucial posterior structure, S5C shows, to the knowledge of the authors, the largest scale applicability. This is even the case if approaches using factored posteriors are included. Notably there is also little reported for very large hidden dimensions for MAP based or deterministic approaches (compare, e.g., [5]), although scalability should be less of an issue. In general it may well be that a method is scalable to larger than the reported latent spaces but that such increases do not result in functional benefits.

For probabilistic approaches, the requirement for approximations with high accuracy have been identified also in other very promising work [30, 31] which uses different approaches that were, so far, applied to much smaller scales. For the select-and-sample method and the spike-and-slab sparse coding model, the high-dimensional applicability means that this or similar approaches are a promising candidate for models such as DBNs, SBNs or CNNs because of their close relation to spike-and-slab models and their typically similarly large scale settings. Here we have studied an application of S5C to standard image patches, primarily to demonstrate scalability. The obtained non-standard generative fields may by themselves, however, be of relevance for V1 encoding (Fig. 2) and they show that spike-and-slab models may be very suitable generalized V1 models. From a probabilistic view on neural processing, the accuracy that can be provided by select-and-sample inference is hereby very desirable and is consistent, e.g., with sampling-based interpretations of neural variability [32]. Here we have shown that such probabilistic approximations are also functionally competitive and scalable to very large hidden dimensions.

**Acknowledgements.** We thank E. Guiraud for help with Alg. 1 (illustration) and acknowledge funding by the DFG: Cluster of Excellence EXC 1077/1 (Hearing4all) and grant LU 1196/5-1.

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
