[Supplementary Material · SheikhLucke-Supplement.pdf]

# Appendix

## Application to Natural Image Patches

Figure 4: Image of all generative fields after S5C was applied with $H = 10\,000$. Latent components $\vec{W}_h$ learned by the S5C algorithm. The components are sorted from left to right and top to bottom with respect to their posterior probabilities averaged over data points.

Figure 5: 400 of the 10 000 generative fields randomly selected for increase readability. Out of $H = 10\,000$ learned components, we pick every $4^{th}$ component for display starting with the most frequently activated fields. Here we show the first 400 such fields.