[Reviews · NeurIPS 2016]

Reviewer 1

Summary

Proposes a new method for estimating sparse coding models with a really large number of latent variables, based on the select-and-sample method.

Qualitative Assessment

This is an application of the select-and-sample method in ref [13] on sparse coding models. In my view, it is quite a simple extension. On the positive side, the section on V1 modelling does give some novel results, although it may be reasonable to doubt that they could be some form of overlearning, and the analysis of not too deep. One of the weakest points here is the presentation. For example, the introduction contains a lot of inaccuracies, e.g. the terms "explaining-away", "plasticity" are used in very approximate ways, ICA is claimed not be "non-probabilistic", etc.

Confidence in this Review

1-Less confident (might not have understood significant parts)


Reviewer 2

Summary

The paper considers the problem of sparse coding, also known as dictionary learning. It is here expressed as a maximum likelihood problem where one wishes to fit to some training data the parameters of a Bernoulli-Gaussian (“spike and slab”) prior on a sparse latent variable of (potentially large) dimension H, the components W of a dictionary, and the variances of the observation noise. To circumvent the intractability of the problem for large H, a technique is proposed which combines ideas from iterative hard thresholding (via a “selection function” which actually keeps the H’ < < H components most correlated to the observation given the current parameters) and expectation maximization with Gibbs sampling. Numerical experiments show that, with an appropriate choice of H’, the approach behaves well both in controlled synthetic experiments (in low dimension) and in one denoising experiments with large H. Further, a discussion of the shape and nature of the learned dictionary atoms is provided.

Qualitative Assessment

Overall this is an interesting and fairly clear algorithmic and experimental paper paper presenting an approximate EM technique for dictionary learning in a Bernoulli-Gaussian setting. The main idea is that EM requires sampling to approximate certain intractable expectations/integrals, and that such sampling is inaccurate or difficult in high dimensions H. It is proposed to perform sampling only after a selection step which consists in finding a subspace (dependent on the considered training vector as well as the currently estimated parameters of the model) of intermediate dimension H’ << H. As far as I could check the selection step, expressed implicitly in (10)-(11) corresponds to computing certain weighted correlations of the training sample y_n with columns of the current estimate of the dictionary, and to keep the H’ largest correlations. This essentially boils down to one step of hard thresholding, reminiscent of certain recent techniques for dictionary learning (see, e.g., recent work of K. Schnass such as [A]). Numerical experiments on synthetic data show state of the art performance of the proposed approach. An discussion on the shape of the atoms learned on the “house” image is provided. On the one hand this is interesting and intriguing, as one can see for example some “curved” atoms similar to those appearing in the work of Jost et al [B], and I am curious see if the approach can be extended to factor invariances (translation, rotation) in the learning stage. On the other hand, it is a bit of a pity that the experiment is only conducted on a single image, since this seriously limits the interpretability of the results. The paper is concluded by the potentially interesting Figure 3 which aims at presenting a synthetic view on existing algorithms based on two axes: “accuracy of posterior approximation” and “latent dimension” (a measure of scalability). I welcome this type of synthetic representations in principle. However I am a bit bothered by the fact that the positioning on the axes (especially the horizontal one) seem to be subjective rather than objective. It would be really interesting to draw such a picture, but one would need to agree for example on the notion of “accuracy of posterior approximation”. One way to do it may be to conduct large-scale empirical simulations and use agnostic performance measures. Since the proposed technique is claimed to be scalable, I would have expected some mention of computation times besides pure log-likelihood figures . Please carefully revise the english writing, e.g.: the sentence line 21-22 has two verbs, line 43 “to in general study”, line 80 “which have in similar forms be derived …”line 181 “implementation following and execute”, line 206 “obtaine”,line 244 “the for MAP based approaches required choice”. [A] Schnass, K. (2014, January 24). Local Identification of Overcomplete Dictionaries. arXiv.org. [B] Jost, P., Vandergheynst, P., Lesage, S., & Gribonval, R. (2006). MoTIF : an Efficient Algorithm for Learning Translation Invariant Dictionaries (Vol. 5, pp. V–V). Presented at the Acoustics, Speech and Signal Processing, 2006. ICASSP 2006. IEEE International Conference on, Toulouse, France: IEEE. http://doi.org/10.1109/ICASSP.2006.1661411

Confidence in this Review

2-Confident (read it all; understood it all reasonably well)


Reviewer 3

Summary

This paper presents a new sparse coding algorithm that blends a probabilistic model, including a spike-and-slab prior, with a Gibbs sampler operating in a reduced latent subspace. This allows for scale scalability up to at least 10^4 dimensional sparse codes as demonstrated via empirical tests, unlike many other sampling-based methods.

Qualitative Assessment

While I appreciate the concerted effort to produce scalable probabilistic solutions, from a novelty standpoint my feeling is that this paper is somewhat limited. Reference [13] already applies a select-and-sample model for analogous neural inference using binary latent variables, and the proposed algorithm extends this to spike-and-slab priors. However, the core concept, of reducing inference to a truncated space for tractable sparse coding, has frequently been applied with spike-and-slab priors for related purposes in the literature (see [7] and references within). Moreover, the selection strategy itself is quite heuristic, and involves the element-wise selection of the H' largest values from a set of degenerate distributions produced from a single nonzero component (see eq. (10) and below). Such greedy dimensionality reductions are commonplace, and many common sparse estimation approaches like iterative hard-thresholding are predicated upon them. Regardless, the proposed pipeline is still largely the adaptation of [7] and [13] to include a Gibbs sampler for computing reduced posteriors in narrow, salient subspaces, and no rigorous convergence analysis is provided. Additionally, for the large-scale empirical tests, the major selling point of the algorithm, there are only qualitative comparisons in terms of visual inspection of receptive fields. Moreover, the MAP approach from [1] (and presumably some others like it) are disparaged for their requirement of a sparsity tuning parameter. But the proposed approach has its own free parameters, like the truncation parameter H' or the number of samples M (... and presumably others?). And reference [1] is now quite old anyway, and there exist many newer penalized regression approaches such as the square-root Lasso that are designed to require only modest tuning. In general though, for such a complex, heuristic approach, it remains unclear to me what real practical advantage exists over much simpler optimization-based approaches emerging from convex analysis and other fields (e.g., reference [5] or Mairal et al., JMLR, 2010) which can truly scale to massive sizes (e.g., millions of variables). Finally, there is considerable emphasis in the paper on the importance of estimating posterior uncertainty. In fact, a similar argument is often made in related contexts. However, in the only quantitative comparison presented, a simple toy denoising problem, much simpler approaches with presumably worse posterior fidelity perform just as well as the proposed method. Admittedly this may be an overly ambitious request, but it would be interesting to see a practical case where better posterior estimates clearly translate into a demonstrable advantage. At the very least, this type of denoising problem no longer seems like a sufficient platform for differentiating practically important performance. I must wholeheartedly admit however, that for computational neuroscience modeling and neural inference, these benchmarks may still be relevant. But I am definitely not up to date on such things.

Confidence in this Review

2-Confident (read it all; understood it all reasonably well)


Reviewer 4

Summary

The paper describes an approximate EM algorithm approach to solving the spike and slab (probabilistic) sparse coding model. They argue that this approach is more accurate than factorised (varational) approaches, since they are able to account for complicated multimodal posteriors with multiple interactions between latent variables. They show that their method is competitive with existing approaches whilst being scalable.

Qualitative Assessment

The paper is generally well written and the approach is appealing. The basic ingredients are to combine the select and sample approach with a spike and slab sparse coding model. However, it seems to me that there doesn't seem to be that much new over the 2011 method of Shelton et al [13], which was also applied to the Sparse Coding problem. I realise that their method was for binary latent variables, but the extension here appears to be only the pointwise multiplication of those binary variables with Gaussian latents. Intuitively it "feels" like the selection part should be essentially the same, with a different sampling part. Given the similarities, this method should also be included in comparisons. The authors should more clearly explain how their method differs. It's not clear to me how figure 3 was produced - is it based on empirical results? Is S3C really more efficient that some of the optimisation based approaches? What are "Coates" and "Li" - they don't seem to be cited? Is there any meaning of the positioning on the x-axis, apart from the categorisation? If so how can this be measured? Minor comments: - The only reference to the fact that they've called the algorithm S5C seems to be in the title of Algorithm 1, and then it's used throughout. - Acronyms should be expanded on first use: MAP, ICA, K-SVD, SBN, MCMC, PSNR, GSC-ET - The full model is not written as a single statement or shown as a graphical model - L70 move "where $\Sigma = ..." to after the definition of all model parameters - L72 \Phi should be defined in a similar way to \Sigma - L76 \langle f(\vec{s} \rangle_n say what this is - L77 the joint of which distributions? this could be made clearer - L80 be derived -> have been derived - L99 withing -> within - eq6 is m the number of samples? - L111 extra right parenthesis after \Reals^{|\mathcal{I}|} - L167 generate -> generated - L117 was drawn -> were drawn - L178 why this choice of \pi values? - L206 obtaine -> obatained - L244 can at least avoid the ... ?

Confidence in this Review

2-Confident (read it all; understood it all reasonably well)


Reviewer 5

Summary

Overall, this paper is good, the idea is described clearly, but the innovation is not enough, and the effect is not obvious.

Qualitative Assessment

The innovation is not enough, compared with the other algorithms, there is no obvious improvement. In many situations, the effect of this paper is less than other algorithms.

Confidence in this Review

2-Confident (read it all; understood it all reasonably well)


Reviewer 6

Summary

This paper scales the spike-and-slab sparse coding for high-dimensional datasets by an approximate inference with select-and-sample. The key extension of this work is the deliverance from the complex posterior limited by the dimension of the index subset. The methodological novelty is clear, but the experimental analysis still needs some improvements.

Qualitative Assessment

1. Be careful with some abbreviations. There's no definition of PSNRs. 2. The paper seems to be an extension of paper [7] with Proposition 1, which is fine. But this is certainly not the only method for large scale sparse coding. So it might be more interesting and convincing to include some quantitative comparisons in the V1 encoding application against other largescale sparse coding methods. Now the results are more of a qualitative presentation.

Confidence in this Review

2-Confident (read it all; understood it all reasonably well)